

# The analysis of the structural parameter influences on measurement errors in a binocular 3D reconstruction system: a portable 3D system

Ou Sha[1,2], Hongyu Zhang[1], Jing Bai[1], Yaoyu Zhang[1] and Jianbai Yang[3]

[1] Changchun Institute of Optics, Fine Mechanics and Physics, Chinese Academy of Sciences, Changchun, China
[2] University of Chinese Academy of Sciences, Beijing, China
[3] Harbin Normal University, Harbin, China

## ABSTRACT

This study used an analytical model to investigate the factors that affect the reconstruction accuracy composed of the baseline length, lens focal length, the angle between the optical axis and baseline, and the field of the view angle. Firstly, the theoretical expressions of the above factors and measurement errors are derived based on the binocular three-dimensional reconstruction model. Then, the structural parameters' impact on the error propagation coefficient is analyzed and simulated using MATLAB software. The results show that structural parameters significantly impact the error propagation coefficient, and the reasonable range of structural parameters is pointed out. When the angle between the optical axis of the binocular camera and the baseline is between 30° and 55°, the ratio of the baseline length to the focal length can be reasonably reduced. In addition, using the field angle of the view that does not exceed 20° could reduce the error propagation coefficient. While the angle between the binocular optical axis and the baseline is between 40° and 50°, the reconstruction result has the highest accuracy, changing the angle out of this range will lead to an increase in the reconstruction error. The angle between the binocular optical axis and the baseline changes 30° through 60° leads to the error propagation coefficient being in a lower range. Finally, experimental verification and simulation results show that selecting reasonable structural parameters could significantly reduce measurement errors. This study proposes a model that constructs a binocular three-dimensional reconstruction system with high precision. A portable three-dimensional reconstruction system is built in the article.

## INTRODUCTION

Three-dimensional (3D) reconstructions can be obtained by directly interfering with the environment using light projectors. Active reconstruction systems that include an integrated red, blue, and green wavelengths (RGB) camera are called red, blue green wavelengths, and depth (RGB-D) sensors as both a color and a depth value can be

Corresponding author
Yaoyu Zhang,
zhangyy9785@163.com

associated with each image pixel. 3D reconstruction methods are classified into passive and active. Passive methods do not involve interaction with the object, whereas active methods use contact or a projection of some form of energy onto the object. 3D reconstruction is an essential technical branch of computer vision (*Moqian, Daniel & Kalanit, 2016*) and uses cameras to capture images of target surfaces and determines spatial coordinates based on the image coordinates of feature points and calibration parameters, thereby achieving spatial measurements and topography acquisitions (*James & Lore, 2010*; *Dumane et al., 2014*; *Tang et al., 2018*). This technology has outstanding advantages such as higher accuracy and efficiency, and non-contacted measurements (*Song et al., 2018*; *Xu et al., 2020*; *Qu, Huang & Zhang, 2018*). Among them, the binocular 3D reconstruction technology is the closest implementation method to human vision (*Shen, 2013*), with more stable algorithms and easier implementation characteristics, and is widely used in technical fields such as industrial measurement and virtual reality with immense development potential (*Huang, Kwok & Zhou, 2019*).

The measurement accuracy of a binocular 3D reconstruction is the most important technical indicator, so there have been many studies on high-precision binocular 3D reconstruction. Among them, algorithms mainly focus on camera calibration and feature recognition. *Yu et al. (2020)* implemented high-precision joint calibration of multiple cameras to minimize reprojection errors (*Yu et al., 2020*; *Huo, Li & Yang, 2018*; *Vo et al., 2011*). *Xu et al. (2019)* completed active visual calibration using high-precision 3D feature points and closed-loop operations with a reconstruction accuracy of 0.05 mm. *Liu et al. (2011)* completed remote sensing image matching using an improved Harris corner detection algorithm. *Kumar, Bansal & Saluja (2021)* obtained an efficient feature point recognition method based on Shi Tomasi's corner detection algorithm. *Qu et al. (2021)* calculated and corrected the imaging error of the through-focus scanning microscopic image, and *Jihoon, Seonghyeon & Kwanghee (2018)* corrected the image distortion of the Kinect projection system, indicating that the manufacturing accuracy of imaging devices, pixel digitization, residual distortion, and other factors in the reconstruction process introduce imaging errors, which have a significant impact on error measurements.

*Guo, Yao & Di (2006)* optimized the structural parameters of the binocular vision system using a parallel structure to improve measurement accuracy. Therefore, research shows that reasonable structural parameters could improve the resolution and measurement accuracy of the system (*Long, Zhengxu & Yiqi, 2008*). The main structural parameters include the length of the binocular baseline, the angle between the optical axis of the binocular camera and the baseline, and the lens focal length (*Liu & Zhi, 2014*). In addition, structural deformation errors may occur due to structural changes caused by gravity, inertial forces, vibration, and temperature changes (*Shao, Zhang & Jia, 2021*; *Lei et al., 2018*). The literature shows that the influence of structural parameters on error measurements is one of the most important factors among many influencing factors. The angle between the optical axis of the camera and the baseline: is α1 and α2, the baseline length: is L, the lens focal length: is f1 and f2, the field of view angle: is v1 and v2, and h1 and h2.

More up-to-date research regarding error propagation and accuracy improvement covering several distinct fields can be found in *Lu et al. (2021)*, *Gai, Da & Dai (2018)*, *Hao et al. (2020)*, *Jingtian et al. (2021)*, *Zhao, Su & Chen (2018)*.

This article aims to study the influence of structural parameters on measurement errors in a binocular 3D reconstruction system and to provide a reasonable range of values for each structural parameter. Moreover, this article aims to establish an analytical relationship between structural parameters and error propagation coefficients using the analytical model. The ideal ranges for structural parameters guided by reducing error propagation coefficients are obtained.

Firstly, a binocular 3D reconstruction model is established based on the principle of triangulation, and the coordinate expression of the target point is established using analytical methods. Then, the theoretical expression of the measurement error propagation coefficient for the system is derived. MATLAB software (Camera Calibration toolbox) is used to simulate and analyze the relationship between the structural parameters and the coefficient of the measurement error propagation to obtain the impact characteristics of the structural parameters on the measurement error magnification and to seek the range of the structural parameters. Finally, theoretical analysis and simulation results are verified through experiments.

The rest of the article is structured as follows. In "Error Analysis Model for the Binocular 3D Reconstruction", an error analysis model for a binocular 3D reconstruction is established based on the principle of triangulation, and the theoretical expression of measurement error magnification is obtained using analytical methods. In "The Derivation and Analysis of Structural Parameters based on Simulation", MATLAB software is used to simulate and analyze the magnification of measurement errors, and the influence characteristics of various structural parameters on the error magnification are obtained. In "Experimentations and Results", multiple experiments were conducted to verify the theoretical calculation and the simulation results. "Discussion and Conclusion" concludes the research.

# ERROR ANALYSIS MODEL FOR THE BINOCULAR 3D RECONSTRUCTION

## Structural parameter design and error analysis model for binocular reconstruction

The structural parameters of a binocular 3D reconstruction system include the angle between the optical axis of the camera and the baseline, the baseline length, the lens focal length, and the field of view angle. Structural parameters are the basic parameters for constructing a measurement system and are the main factors that affect the reconstruction accuracy and the size of the common field of view. The structural parameters are both independent of each other and mutually affect and restrict each other (*Kim & Tzempelikos, 2021*). As shown in Fig. 1, it is a model analyzing errors for a binocular reconstruction system. The optical axis of the binocular camera is located in plane M, and the included

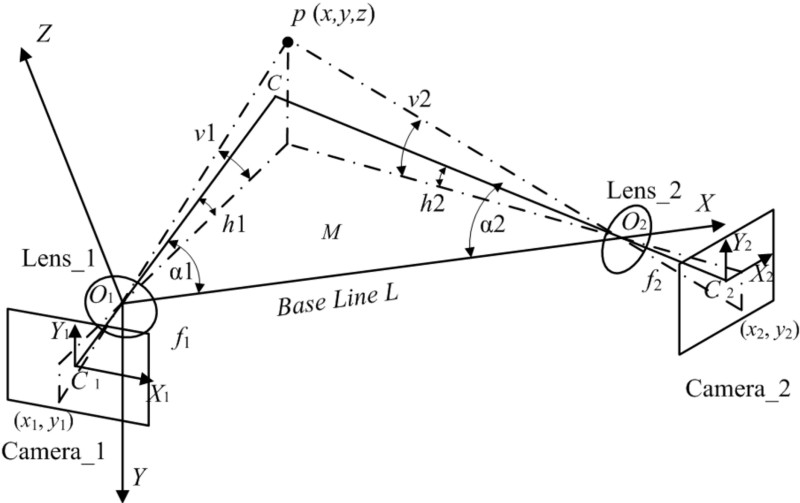

**Figure 1 Measuring binocular reconstruction system's accuracy analysis model.**

angles with the baseline are α1 and α2, respectively, and intersect at point C, with a baseline length of L, the same parameters for binocular cameras, and lens focal lengths of f1 and f2 (*Zhou et al., 2019*). The point p (x, y, z) to be measured in space is imaged on the left and right eye cameras, with image coordinates of (x1, y1) and (x2, y2), respectively. To facilitate analysis, the optical center O1 of the left eye camera lens is used as the coordinate origin O, with the baseline as the X-axis, the positive direction points to the optical center O2 of the right eye camera, the Y-axis is perpendicular to the plane M and points downward. The right-hand rule determines the direction of the Z-axis. The included angles between the projection point p on the plane M and the optical axis are h1 and h2, respectively. The projection point distance from point p to plane M relative to the height angle of the binocular camera is represented by v1 and v2. According to the ideal imaging model of the camera, the spatial coordinates of point p are shown in Eq. (1).

According to the principles of camera imaging and triangulation (*Hua et al., 2022*), the spatial coordinates (x, y, z) of a point p are calculated using the image coordinates (x1, y1) and (x2, y2) of the point p in the binocular image by ignoring the influence of lens distortion, as shown in Eq. (1). The spatial coordinates of the point p could be solved based on the structural parameters of the system under ideal camera imaging conditions. Therefore, the spatial coordinates of the point p could be considered a multivariate function containing structural parameters, as shown in Eq. (2).

However, considering the impact of errors introduced during imaging and calibration on the spatial coordinates of the point p, as shown in Eq. (3). The spatial coordinate error source of the point p includes the calibration error of five structural parameters ΔL, Δα1, Δα2, Δf1, Δf2, and the measurement errors of the four measured quantities Δx1, Δy1, Δx2, Δy2 from Eqs. (2) and (3). The contributions of these error sources to each coordinate component can be written in a partial differential equation, as shown in Eq. (4). The error

values for each element are obtained by multiplying and summing each error source and the corresponding partial derivative Pij.

$$
\begin{cases}
x = \dfrac{L \cdot \tan \theta_2}{\tan \theta_1 + \tan \theta_2} \\
y = y_1 \cdot \dfrac{x \cos h_1}{f_1 \cos \theta_1} \\
z = \dfrac{L}{\cot \theta_1 + \cot \theta_2} \\
h_1 = \arctan(x_1/f_1) \\
h_2 = \arctan(x_2/f_2) \\
v_1 = \arctan(y_1 \cos h_1/f_1) \\
v_2 = \arctan(y_2 \cos h_2/f_2) \\
\theta_1 = \alpha_1 + h_1 \\
\theta_2 = \alpha_2 + h_2
\end{cases}
\tag{1}
$$

$$
(x, y, z) = P(L, \alpha_1, \alpha_2, f_1, f_2, x_1, y_1, x_2, y_2)
\tag{2}
$$

$$
\Delta P = \sqrt{\Delta x^2 + \Delta y^2 + \Delta z^2}
\tag{3}
$$

$$
\begin{bmatrix} \Delta x \\ \Delta y \\ \Delta z \end{bmatrix} =
\begin{bmatrix}
P_{11} & P_{12} & \cdots & P_{18} & P_{19} \\
P_{21} & P_{22} & \cdots & P_{28} & P_{29} \\
P_{31} & P_{32} & \cdots & P_{38} & P_{39}
\end{bmatrix}
\begin{bmatrix} \Delta L \\ \Delta \alpha_1 \\ \vdots \\ \Delta x_2 \\ \Delta y_2 \end{bmatrix}
\tag{4}
$$

where $\Delta L$, $\Delta \alpha 1$, $\Delta \alpha 2$, $\Delta f 1$, $\Delta f 2$ are the calibration errors of structural parameters, $\Delta x 1$, $\Delta y 1$, $\Delta x 2$, $\Delta y 2$ are the measurement errors of image coordinates, and Pij is the partial derivative of the coordinate error of the point p for each structural parameter, and each partial derivative constitutes a partial derivative matrix P.

Pij is the amplification factor of the contribution value of the error source to the coordinate error of the point p. To minimize the positional error of the point p as much as possible, it is hoped that under the same calibration error and the same measurement error conditions, each element of the partial derivative matrix P will have as small a value as possible to reduce its amplification effect on the error source. The partial derivative matrix can also be called an error propagation coefficient.

This article aims to establish an analytical relationship between structural parameters and error propagation coefficients using the analytical model shown in Fig. 1 and to obtain an ideal range for structural parameters guided by reducing error propagation coefficients. Generally, the calibration error of structural parameters mainly results from the calibration error of the calibrator and the error of the calibration algorithm. After the beam adjustment operation, the calibration error is further reduced, and the calibration accuracy would be higher; the measurement errors of the measured quantities mainly result from lens distortion and errors generated during digitization. This error is directly brought into the calculation, which is more significant than the calibration error.

# THE DERIVATION AND ANALYSIS OF STRUCTURAL PARAMETERS BASED ON SIMULATION

## Influence of the system structural parameters on the error propagation coefficient

The structural parameters of a binocular reconstruction system determine the measurement triangle's basic shape during system design. The angle between the optical axis of a binocular camera and the baseline determines the viewing direction of the target under test, which directly impacts the reconstruction effect and needs to be prioritized in practice. Subsequently, it is necessary to select the lens focal length and determine the baseline length, which affects the camera's field of view angle and the depth of the field. The view angle from the target point to the camera's optical axis also affects the measurement accuracy. Each parameter will be analyzed in the next subsection. All equations presented in the subsequent sections are derived from *Yang et al. (2018)*, *Fan, Cheng & Fu (2015)*, *Hu et al. (2020)*.

## The influence of an angle between the binocular optical axis and the baseline on the error propagation coefficient

(1) Influence on the measured error quantity

We establish a target point coordinate expression composed of image coordinates $(x1, y1)$, $(x2, y2)$, and structural parameters according to Eq. (1), wherein the image coordinates are referred to as the measured quantity. To analyze the impact of the measured error quantities on the spatial coordinate errors of the target point, the propagation coefficients of measured error quantities in each coordinate component are obtained using Eqs. (1)–(4).

$$\begin{cases} \dfrac{\partial x}{\partial x_1} = -\dfrac{L \sin 2\theta_2 \cos^2 h_1}{2 f_1 K^2} \\[2mm] \dfrac{\partial x}{\partial x_2} = -\dfrac{L \sin 2\theta_1 \cos^2 h_2}{2 f_2 K^2} \\[2mm] \dfrac{\partial y}{\partial y_1} = \dfrac{L \cos h_1 \sin \theta_2}{f_1 K} \\[2mm] \dfrac{\partial y}{\partial y_2} = \dfrac{L \cos h_2 \sin \theta_1}{f_2 K} \\[2mm] \dfrac{\partial y}{\partial x_1} = -\dfrac{L \tan v_1 \cos h_1 \cos(\alpha_1 + \theta_2) \sin \theta_2}{f_1 K^2} \\[2mm] \dfrac{\partial y}{\partial x_2} = \dfrac{L \tan v_1 \sin \theta_1 \cos^2 h_2}{f_2 K^2} \\[2mm] \dfrac{\partial z}{\partial x_1} = \dfrac{L \cos^2 h_1 \sin^2 \theta_2}{f_1 K^2} \\[2mm] \dfrac{\partial z}{\partial x_2} = \dfrac{L \cos^2 h_2 \sin^2 \theta_1}{f_2 K^2} \end{cases} \tag{5}$$

where $K = \sin(\alpha_1 + h_1 + \alpha_2 + h_2)$.

To facilitate analysis, assume that the image coordinates (x1, y1) and (x2, y2) have equal errors. According to Eq. (5), the relationship between the error propagation coefficient of the measured quantity regarding the position of the point p and the included angle of the optical axis is obtained as follows:

$$P_{xy} = \sqrt{\left(\frac{\partial x}{\partial x_1}\right)^2 + \ldots + \left(\frac{\partial y}{\partial y_1}\right)^2 + \left(\frac{\partial y}{\partial y_2}\right)^2 + \left(\frac{\partial z}{\partial x_1}\right)^2 + \left(\frac{\partial z}{\partial x_2}\right)^2} \qquad (6)$$

The complete form of Eq. (6) is relatively complex. To accurately reflect its change rule, a 3D image is drawn using MATLAB software. The angle between the binocular optical axis and the baseline varies from 30° to 60°, and the relationship between the error propagation coefficient and α1 and α2 is shown in Fig. 2A. The simulation data shows that the value of the included angles α1 and α2 has a significant impact on the error propagation coefficient. The error propagation coefficient has a significant upward trend as the two included angles increase. It is worth noting that at locations where the two included angles are equal, the error propagation coefficient is smaller than at locations where the two included angles are unequal. Therefore, the angle between the binocular optical axis and the baseline should be as close as possible, which is beneficial to reducing the error propagation coefficient.

To further analyze the influence of the angle between the binoculars on the error propagation coefficient, the variation rule of the error propagation coefficient at the crossover point of the binocular optical axis is investigated, *i.e.*, h1 = h2 = v2 = v2 = 0, the simplified error propagation coefficient is defined by

$$\frac{\partial p}{\partial \alpha} = \frac{L \sin \alpha}{2 \sin^2(2\alpha)} \qquad (7)$$

A two-dimensional image of this function is drawn in MATLAB software. When the angle between the binocular optical axis and the baseline varies from 0° to 90°, the relationship between the single error propagation coefficient and the angle is shown in Fig. 2B. The simulation data shows that when the angle between the optical axis and the baseline is within the range close to 0°, the error propagation coefficient presents an increasing trend since the angle of the optical axis decrease. While the angle is within the range changing between 25° to 55°, the error propagation coefficient would be small. When the angle is greater than 60°, the error propagation coefficient increases sharply, significantly amplifying the measured error quantity. Suggested that when designing a binocular reconstruction system, the angle between the optical axis of the binocular camera and the baseline should be between 25° and 60° to reduce the error propagation coefficient. At the same time, a symmetrical arrangement with equal left and right angles should be adopted.

(2) Influence on the calibration error

Although there are many high-precision calibration methods, such as Tsai's subsection calibration, Zhang's calibration method, and beam adjustment optimization, calibration

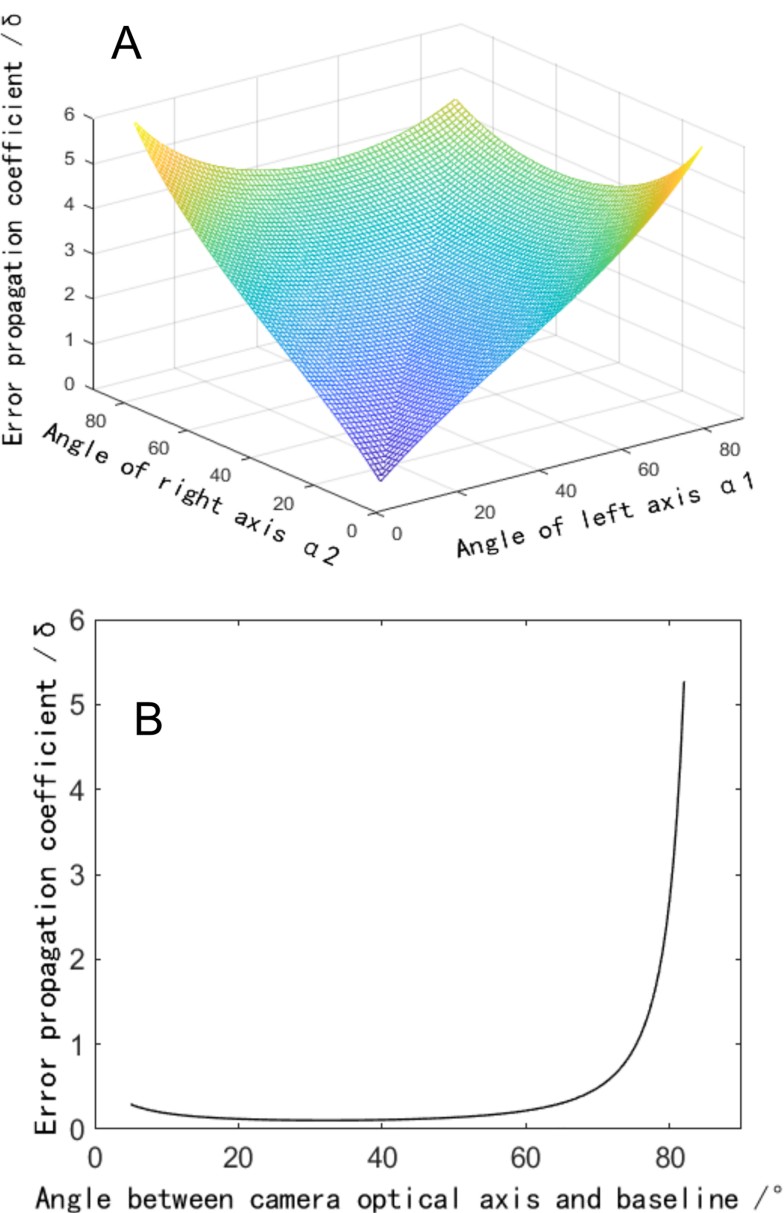

**Figure 2 Influence of the angle between the binocular optical axis and the baseline on the error propagation coefficient.** (A) Influence distribution of the angle between the binocular optical axis and the baseline on the error propagation coefficient. (B) The influence of the angle between the optical axis and the baseline on the error propagation coefficient in a symmetrical arrangement.

errors still exist inevitably due to the influence of algorithms or hardware performances. These errors are accompanied by calibration parameters to participate in the operation, which introduces errors into the final measurements. According to Eqs. (1) and (2), the

relationship between the error propagation coefficient and the included angle of the optical axis is obtained as follows:

$$
\begin{cases}
\dfrac{\partial x}{\partial \alpha_1} = -\dfrac{L \sin \theta_2 \cos \theta_2}{K^2} \\
\dfrac{\partial x}{\partial \alpha_2} = \dfrac{L \sin \theta_1 \cos \theta_1}{K^2} \\
\dfrac{\partial y}{\partial \alpha_1} = -\dfrac{L \sin \theta_2 \tan h_1 \cos(\theta_1 + \theta_2)}{K^2} \\
\dfrac{\partial y}{\partial \alpha_2} = \dfrac{L \tan h_1 \sin \theta_1}{K^2} \\
\dfrac{\partial z}{\partial \alpha_1} = \dfrac{L \sin^2 \theta_2}{K^2} \\
\dfrac{\partial z}{\partial \alpha_2} = \dfrac{L \sin^2 \theta_1}{K^2}
\end{cases}
\tag{8}
$$

The error propagation coefficient of the coordinates of the target point could be expressed as the sum of the terms in Eq. (8), which is defined by.

$$
P_\alpha = \sqrt{\left(\dfrac{\partial x}{\partial \alpha_1}\right)^2 + \left(\dfrac{\partial x}{\partial \alpha_2}\right)^2 + \left(\dfrac{\partial y}{\partial \alpha_1}\right)^2 + \left(\dfrac{\partial y}{\partial \alpha_2}\right)^2 + \left(\dfrac{\partial x}{\partial \alpha_1}\right)^2 + \left(\dfrac{\partial y}{\partial \alpha_2}\right)^2}
\tag{9}
$$

Figure 3A depicts that when the included angle between the binocular optical axis and the baseline changes from 10° to 80°, the total error propagation coefficient tends to increase at both ends and remains flat in the middle. At the same time, it shows obvious symmetry, that is, the value of the included angles α1 and α2 are exchanged. Thus, the error propagation coefficient remains unchanged. If the symmetrical plane of the image extending to both sides, the vertical coordinate of the surface changes little. It can be considered that if the sum of the included angles $\alpha_1$ and $\alpha_2$ would be a constant, the error propagation coefficient could not change obviously when the two included angles change. Therefore, we further analyze the symmetrical plane position of the curved surface and draw the image as shown in Fig. 3B. When the included angle would be small, the error component of the x direction increases significantly, which makes the total error increase. When the included angle increases to 60°, the error component of the z direction increases significantly, which also increases the total error. In the whole range, the error component of the y direction is small and the changing trend is not obvious. These three items conform to the geometric characteristics of the triangulation principle. So, when the angle between the binocular optical axis and the baseline changes 30° through 60°, the error propagation coefficient would be in a low range, which could reduce the measurement error caused by calibration error. Therefore, when designing a binocular 3D reconstruction system, the included angle of the binocular is usually consistent, which can not only make greater use of the camera field of view but also reduce the difficulty of the algorithm.

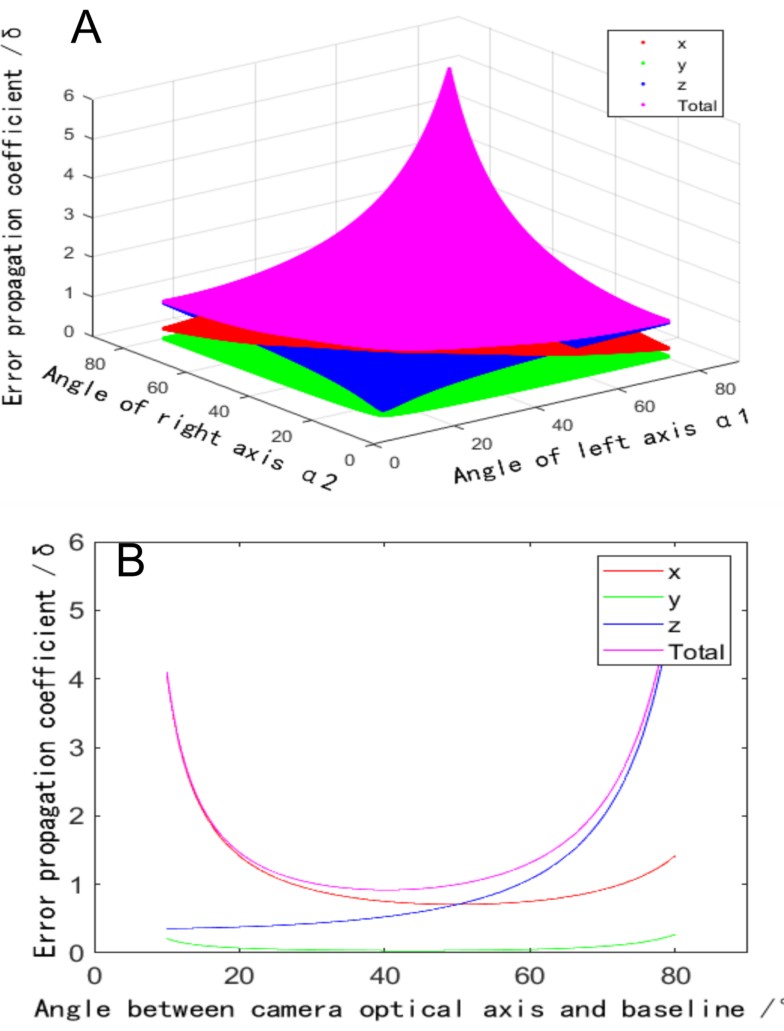

**Figure 3 Influence of the angle between the binocular optical axis and the baseline on the calibration coefficient of the error propagation.** (A) Influence distribution of the angle between the binocular optical axis and the baseline on the calibration coefficient of the error propagation (x is red and y is green). (B) The influence of the angle between the optical axis and the baseline on the calibration coefficient of the error propagation in a symmetrical arrangement.

### Influence of the baseline length on the error propagation coefficient

Equation (4) shows that when the included angle is constant, each error propagation coefficient is proportional to the baseline length $L$, and combined with the ideal imaging model of the camera. $L/f$ is proportional to the magnification of the camera, and $L$ is also proportional to the object distance of the point $p$. Therefore, when other conditions are constant, reducing the baseline length or selecting a lens with a long focal length could reduce the measurement error since a telephoto lens is selected to improve the shooting resolution. However, reducing the magnification also makes the public field of view smaller, which will limit the size of the public field of view of binocular cameras. Therefore,

one of the constraints on the baseline length is the mutual restriction between the measurement accuracy and the field of view (*Hangjun, 2021*).

The second constraint is the depth of field (*Yang et al., 2020*). According to the analysis model in Fig. 1, when the baseline length $L$ is changed and other parameters are unchanged, the sum of the points $C, O_1, O_2$ forms a cluster of similar triangles, and the size of the triangle depends on the baseline length $L$. During reconstruction, the effective acquisition area of a binocular camera is the spatial intersection of the public field of view and the depth of the field, as shown in the green area in Fig. 4A. When the baseline length $L$ decreases, the common field of view moves towards the camera. While the front and back depths of the field of the camera remain unchanged, and the overlapping area of the common field of view and the depth of field is compressed, at this time, the range of the acquisition area is reduced, as shown in the red area in Fig. 4B. When the target exceeds the acquisition area, the imaging clarity decreases, increasing reconstruction error. If the baseline length $L$ is increased, the common field of view moves away from the camera, and the overlapping area with the depth of the field decreases, as shown in the yellow area in Fig. 4C. When the target exceeds the acquisition area, the reconstruction error increases.

## Influence of the focal length on the error propagation coefficient

When other structural parameters remain unchanged, the focal length of the lens determines the depth of the field and the field of view of the camera and has an important influence on the measurement accuracy of the reconstruction system. To accurately quantify the influence of the lens' focal length on measurement error, an expression for the influence of the focal length on error propagation coefficients could be established according to Eqs. (1)–(3):

$$
\begin{cases}
\dfrac{\partial p}{\partial f_1} = \dfrac{c_1 L \sin(\alpha_2 + h_2) \sin h_1}{f_1 K^2} \\[2mm]
\dfrac{\partial p}{\partial f_2} = \dfrac{c_2 L \sin(\alpha_1 + h_1) \sin h_2}{f_2 K^2} \\[2mm]
c_1 = \sqrt{\cos^2 h_1 + \tau_1^2 \sin^2(\alpha_1 + h_1)\left[K\cos\alpha_1 - \cos h_1 \sin(\alpha_2 + h_2) - K\dfrac{\sin(\alpha_1 + h_1)}{\sin h_1}\right]} \\[2mm]
c_2 = \sqrt{\cos^2 h_2 + \tau_2^2 \sin^2(\alpha_2 + h_2)\left[K\cos\alpha_2 - \cos h_2 \sin(\alpha_1 + h_1) - K\dfrac{\sin(\alpha_2 + h_2)}{\sin h_2}\right]} \\[2mm]
\tau_1 = \dfrac{\tan v_1}{\sin(\alpha_1 + h_1)} \\[2mm]
\tau_2 = \dfrac{\tan v_2}{\sin(\alpha_2 + h_2)}
\end{cases}
\tag{10}
$$

Equation (10) shows that when other parameters are constant, the error propagation coefficient is inversely proportional to the focal length, so choosing a lens with a long focal length could reduce the error propagation coefficient. However, through the imaging model of the camera, the smaller the focal length of the lens, the larger the field of view and the larger the public field of view would be. When the focal length increases, the angle of view decreases and the common field of view decreases. The measurement accuracy and

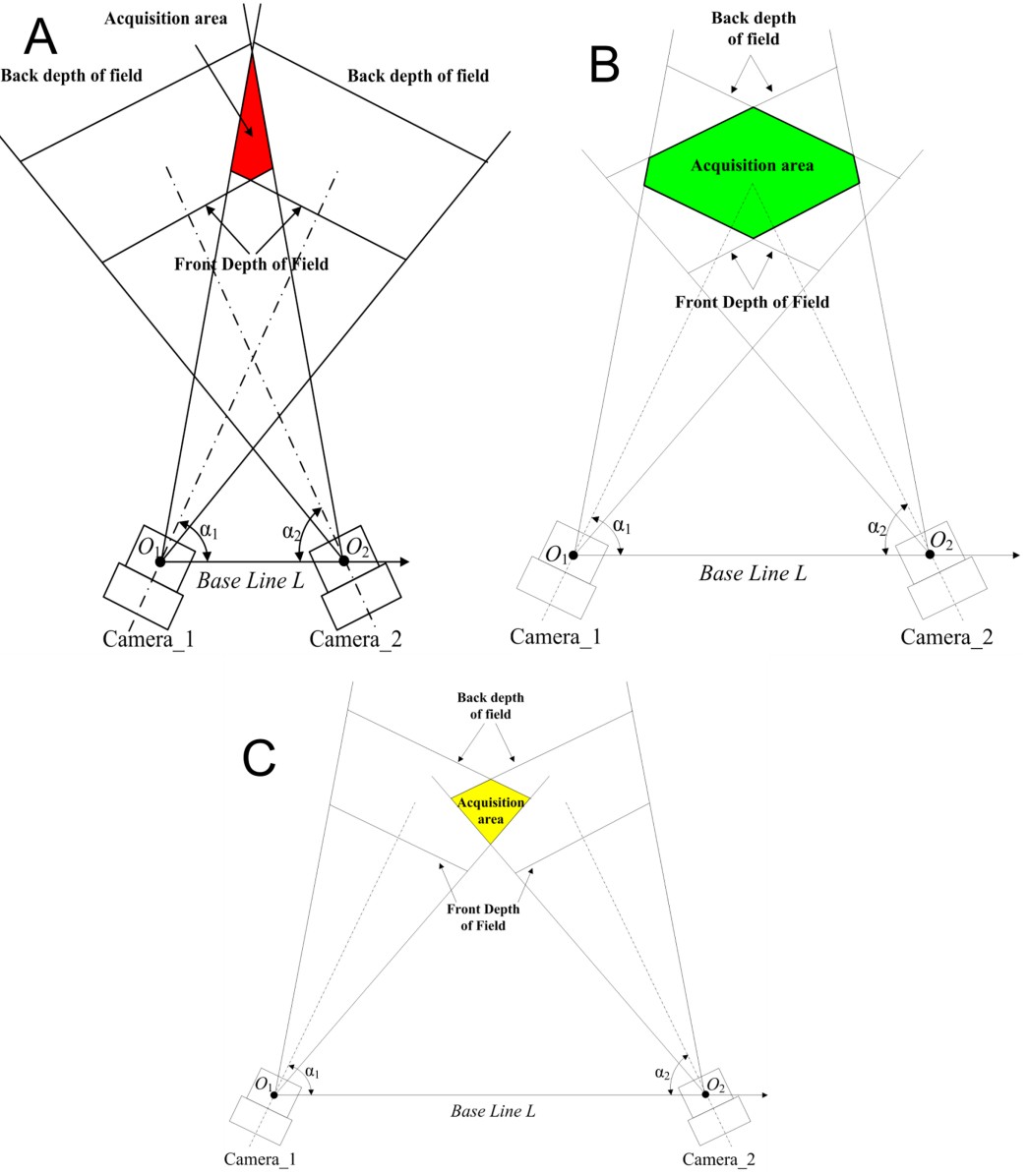

**Figure 4 Influence of baseline distance on the size of the acquisition area.** (A) Acquisition area with moderate baseline length (the colored area green). (B) Acquisition area with reduced baseline length (the colored area red). (C) Acquisition area with increased baseline length.

measurement range are mutually restricted. Therefore, when designing structural parameters, the lens with a long focal length should be selected as far as possible under the condition that meets the field of view, which could thus reduce the reconstruction error.

## Influence of the horizontal field of view angle on the reconstruction error

In Fig. 1, the horizontal viewing angles $h_1, h_2$ from the point $p$ to the optical axis of binocular cameras reflect the projected position of the point $p$ on the plane $M$. When other

conditions are not changed, the relationship between the horizontal field of view angle and error propagation coefficient can be established by changing the size of the horizontal field of view angle:

$$\begin{cases} \dfrac{\partial p}{\partial h_1} = \dfrac{L}{K^2 f} \sqrt{(K^2 + \cos^2 h_1 + g_1^2)\cos^2 h_1 \sin^2(\alpha_2 + h_2)} \\ \dfrac{\partial p}{\partial h_2} = \dfrac{L}{K^2 f} \sqrt{(K^2 + \cos^2 h_2 + g_2^2)\cos^2 h_2 \sin^2(\alpha_1 + h_1)} \\ g_1 = \tau_1[\cos h_1 \sin(\alpha_2 + h_2) + K \cos \alpha_1] \\ g_2 = \tau_2[\cos h_2 \sin(\alpha_1 + h_1) + K \cos \alpha_2] \end{cases} \tag{11}$$

Equation (11) suggests that the distribution influence of the field of view angles $h_1$, $h_2$ on the error propagation coefficient can be obtained. In Fig. 5, $h_1 \in [-30°, 30°]$, $h_2 \in [-30°, 30°]$, the angle between the binocular optical axis and the baseline $\alpha_1 = \alpha_2 = 40°$, $v_1 = v_2 = 30°$ are assumed. Figure 5 depicts that the variation trend of the error propagation coefficient decreases with the reduction of the field of view angle, has a minimum value in the central area, and gradually increases from the center to the periphery. The error propagation coefficient fluctuates slightly in the symmetrical axis region of the field of view angle $h_1 = h_2$. Therefore, the error propagation coefficient caused by the horizontal field of view angle is the smallest at the intersection of binocular optical axes. In the middle vertical plane area of the baseline, the error propagation coefficient caused by the horizontal field angle is smaller than that in other areas.

**Influence of the vertical field of view angle on the reconstruction error**

As shown in Fig. 1, the vertical viewing angle is the included angle connecting the line between the target point $p$ and the camera and the plane $M$, which reflects the height angle of the point $p$ deviating from the plane $M$. When the vertical field of view angle changes, the target point is located in different positions in the field of view, and the error propagation coefficient of the point position concerning the vertical field of view angle $v_1$ and $v_2$ is obtained according to Eq. (1):

$$\begin{cases} \dfrac{\partial y}{\partial v_1} = \sec^2 v_1 \sec(h_1/f_1) \\ \dfrac{\partial y}{\partial v_2} = \sec^2 v_2 \sec(h_2/f_2) \end{cases} \tag{12}$$

Equation (12) suggests that the components of the $X$ and the $Z$ axes of the point p's position are not affected by the vertical field angle, but the $Y$ axis components are affected by the vertical field angle at the same time. Therefore, the error propagation coefficient of the point position error concerning the vertical field angle $v_1$ and $v_2$ is defined by.

$$\frac{\partial p}{\partial v} = \sqrt{\left(\frac{\partial y}{\partial v_1}\right)^2 + \left(\frac{\partial y}{\partial v_2}\right)^2} \tag{13}$$

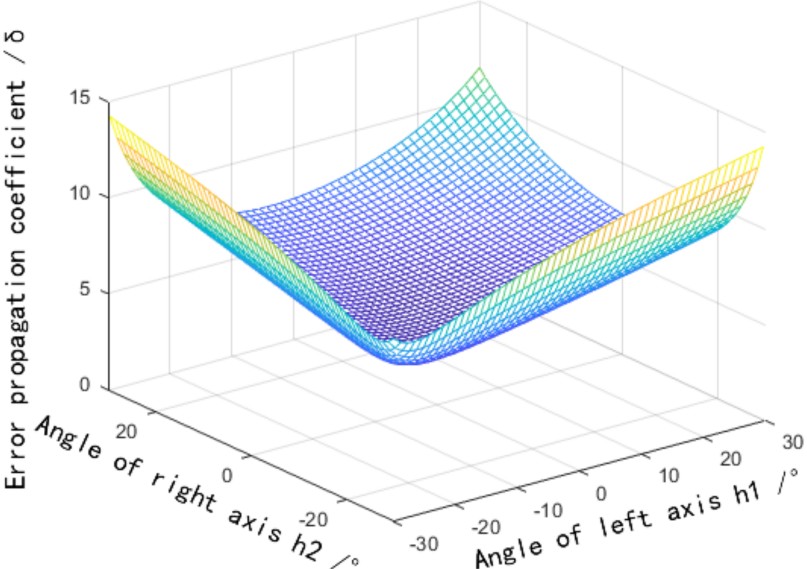

**Figure 5  Influence distribution of a horizontal field angle on the error propagation coefficient.**

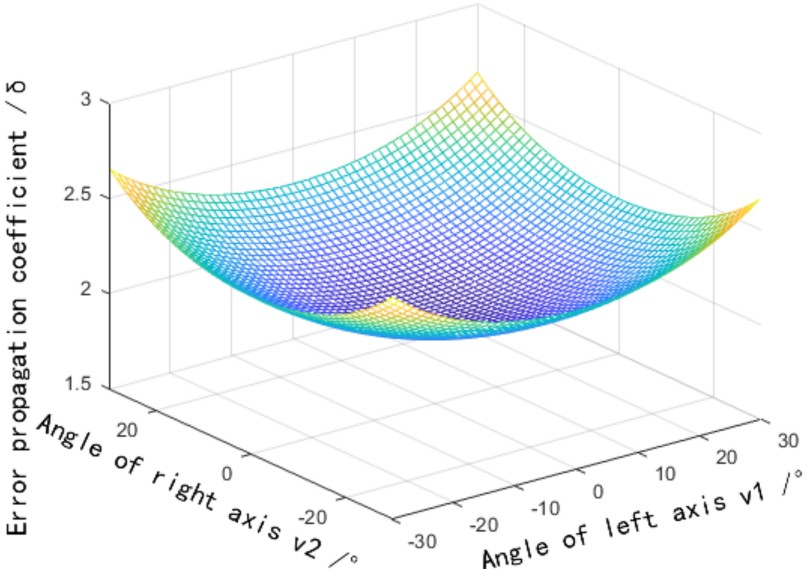

**Figure 6  Influence distribution of the vertical field angle on the error propagation coefficient.**

According to Eq. (13), the distribution of the error measurements varies with the field of view angle established, as shown in Fig. 6. More detailed discussions can be found in *Yang et al. (2018), Fan, Cheng & Fu (2015), Hu et al. (2020)*. The horizontal field of view angle $h_1 = h_2 = 5°$, vertical field of view angle, $v_1 \in [-30°, 30°]$, $v_2 \in [-30°, 30°]$ are assumed. The error propagation coefficient increases with the increase of the vertical field of view angle, and the error caused by the vertical field of view angle is the smallest at the center position $v_1 = v_2 = 0°$, that is, the target point $p$ is located on the plane $M$.

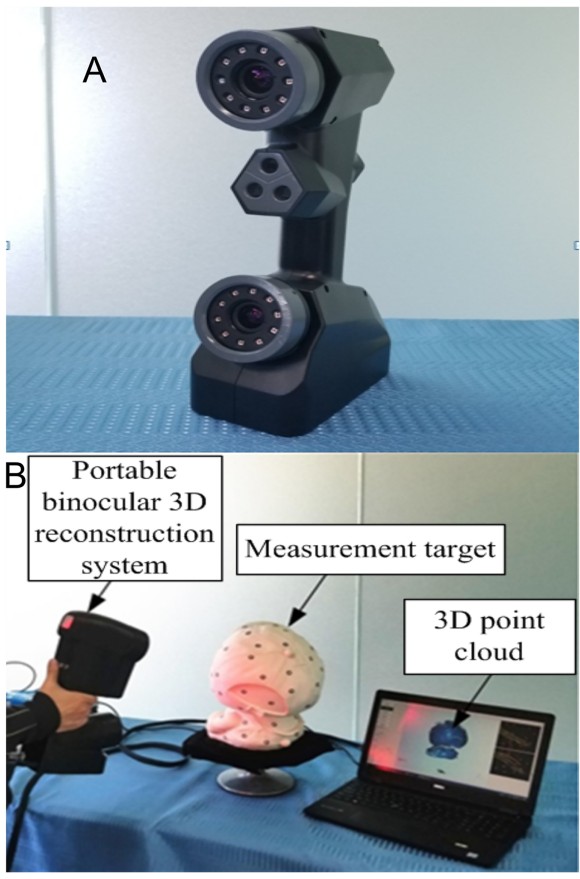

**Figure 7  Portable binocular 3D reconstruction system (taken by the author).** (A) Portable binocular 3D reconstruction system. (B) Status at work.               

## EXPERIMENTATIONS AND RESULTS

To verify the validity of the conducted methods and simulation, a portable binocular 3D reconstruction system as shown in Fig. 7 is built in this article, and two experiments are designed: (1) The influence of the angle between the binocular optical axis and the baseline on the error propagation coefficient. (2) The influence of the horizontal field of view and the vertical field of view on the error propagation coefficient. To ensure the accuracy of the measurement results of the binocular 3D reconstruction system, firstly, the reconstruction system in Fig. 7 is tested, and the standard ceramic balls and right-angle blocks shown in Fig. 8 are selected as the test objects.

### Data collection and the design of the experiment

Two sets of point clouds are obtained by the binocular reconstruction of two test objects, as shown in Fig. 9. Geomagic, a point cloud processing tool, is used to fit the point cloud of the standard ceramic ball in Fig. 9A, and it is found that the diameter of the ball is 50.8234 mm, the accurate value of the diameter provided in the specification of the standard ceramic ball is 50.8011 mm, and the error is 0.0223 mm. At present, the measurement accuracy of domestic mainstream 3D reconstruction systems is 0.03 mm, and the

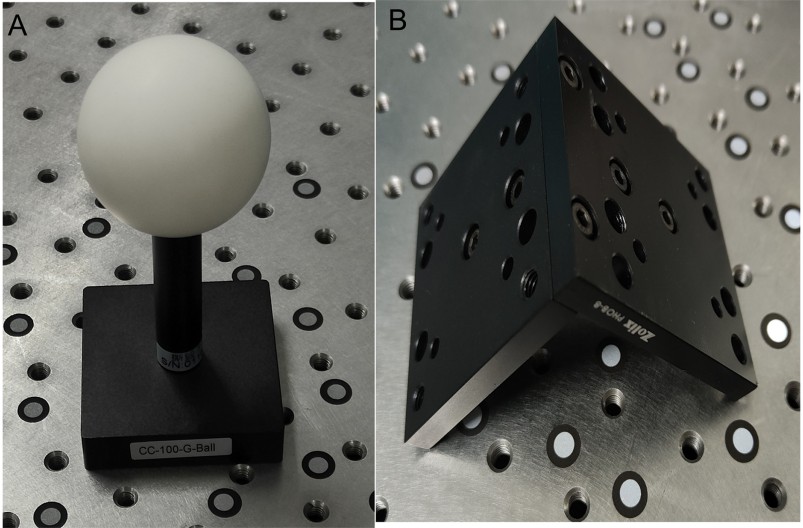

**Figure 8 The test objects (taken by the author).** (A) A standard ceramic ball. (B) Right angle block.

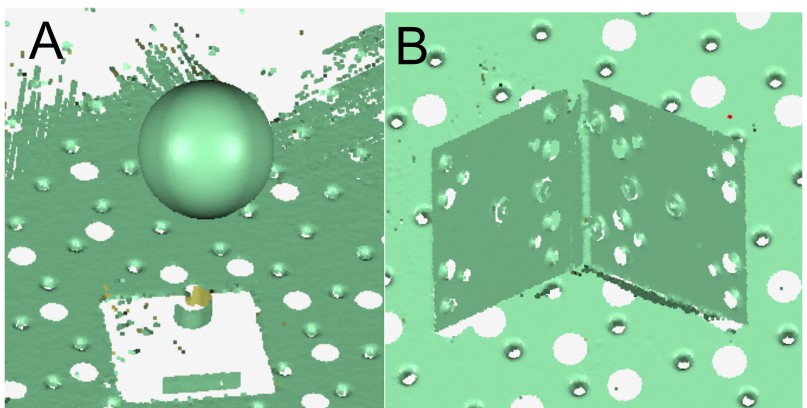

**Figure 9 Reconstruction results.** (A) The point cloud of the standard ceramic sphere (the color is green). (B) The point cloud of the right-angle block (the color is green).

binocular reconstruction system built in this article is slightly higher than similar reconstruction systems in China. The included angle between the two fixed surfaces of the right-angle block is fit to be 90.013°, and the included angle between the two surfaces is 89.986° by using the digital display angle ruler, and the deviation is 0.027°. The measurement results of absolute size and angle show that the measurement accuracy of this reconstruction system can meet the experimental requirements.

## Influence of the angle between the binocular optical axis and baseline on the reconstruction error

Figure 10 shows an experimental device for testing the influence of the included angle between the binocular optical axis and the baseline on the reconstruction error, and its hardware configuration and algorithm are consistent with the reconstruction system in

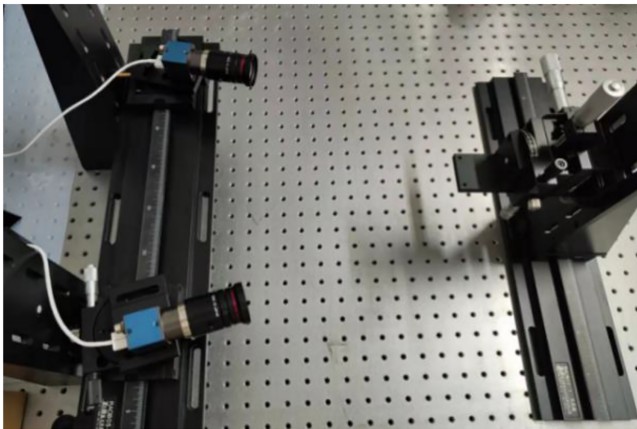

**Figure 10 Experimental device for the influence of the included angle between the binocular optical axis and baseline on the reconstruction error (taken by the author).**

Fig. 7. Specifically, it includes two industrial cameras with a resolution of 2,048 × 1,536 and a pixel size of 3.45 μm, which are symmetrically arranged in the horizontal direction. The cameras are connected to a high-precision turntable and fixed on a guide rail by a slider, and the length of the baseline and the included angle between the optical axis and the baseline can be adjusted by the turntable and the slider. The motion platform is composed of a plurality of precision displacement tables, and the sensitivity is 0.003 mm.

The circular landmark points pasted on the moving platform are used as the target. In this experiment, the motion axis of the motion platform is set in the direction roughly parallel to the coordinate system of the binocular reconstruction system, and the spatial coordinates of the landmark points are collected at multiple positions within the field of view, and the difference between the relative displacement of the landmark points and the displacement of the precision displacement table is regarded as the reconstruction error of the binocular reconstruction system.

The angle between the binocular optical axis and the baseline is adjusted at intervals of 10°, with an adjustment range of 30° through 80°. At the same time, the position of the motion platform is adjusted, so that the motion range of the marker points is near the intersection of binocular optical axes. Therefore, the influence of the field of view angle is reduced. Every time it is adjusted, the binocular camera needs to be calibrated again, so the included angle of the binocular should not be too small. After the calibration is completed, the precision displacement table is moved to make the marker point move along a trajectory like the Chinese character '田', and the precise distance of each intersection point is 10 mm. The position of the collected marker point in each area is shown in Fig. 11. The spatial coordinates of the marker point at nine intersections are measured respectively, and the reconstruction result of the marker point displacement is calculated, as shown in Fig. 12.

Figure 12 shows that when the angle between the binocular optical axis and the baseline is approximately 40°, the average error of the reconstruction results is the smallest, which is only 0.0108 mm. The included angle between the binocular optical axis and the baseline

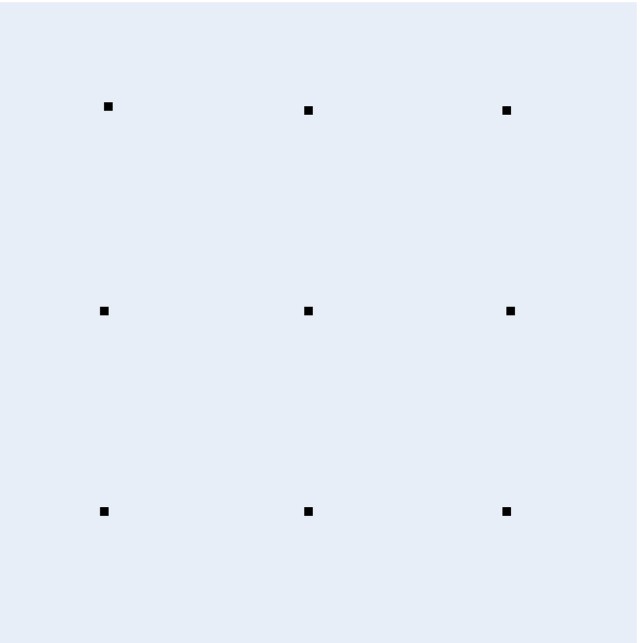

**Figure 11 Location map of marker point acquisition.**

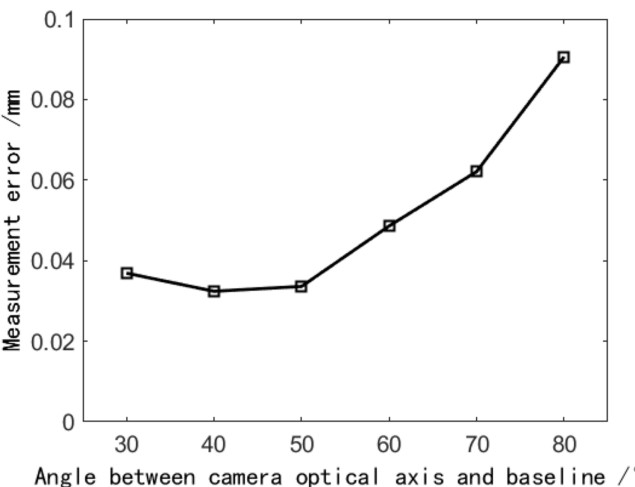

**Figure 12 The effect of the angle between the binocular optical axis and baseline on the reconstruction error.**

varies between 30° through 50°, and the average error of the reconstruction results is relatively small. When the included angle exceeds 50°, the average error of reconstruction results increases with the increase of included angle. This result is consistent with the conclusion of the previous theoretical analysis, which shows its effectiveness. Therefore, when the angle between the binocular optical axis and the baseline is between 40° and 50°, the reconstruction result has the highest accuracy, and increasing or decreasing the angle will lead to an increase in the reconstruction error.

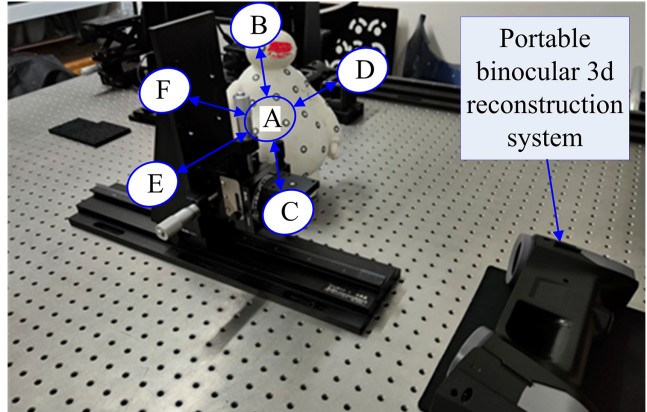

**Figure 13 Experimental device for the influence of horizontal field angle and vertical field angle on the reconstruction error (taken by the author).**

**Table 1 Reconstruction results of landmarks in different field angle regions.**

| Acquisition position | A | B | C | D | E | F |
|---|---|---|---|---|---|---|
| Measurement error | 0.036 | 0.074 | 0.076 | 0.149 | 0.122 | 0.043 |

## Influence of the horizontal and vertical fields of view angles on the reconstruction error

As shown in Fig. 13, to measure the influence of different horizontal and vertical fields of view angles on the reconstruction results, an experimental device was built according to the experimental principle. As there is no need to adjust the included angle and distance of the optical axis of the binocular camera, the portable binocular reconstruction system shown in Fig. 7 was used for the experiment, and it also included a motion platform and circular marker points pasted on the motion platform.

The motion platform is placed in six areas, such as A through F, in the field of view, where A, B, C, D, and E are located on a plane about 250 mm away from the *XOY* surface of the reconstruction system, A is in the center, B–E is located in four directions of A, with a distance of 100 mm, and F is located behind A, with a distance of about 350 mm from the *XOY* surface. The precision displacement table is moved along a trajectory like the Chinese character '田' in six areas, and the spatial coordinates of the mark point are measured at nine intersections, where the precise distance of each intersection is 10 mm, and the collection results are shown in Table 1.

The measurement results in Table 1 show that area A is located in the center of the field of view, and compared with the other five areas, its average measurement error is the smallest, with an absolute error of 0.012 mm. Areas B and C are close to the left and right edges of the field of view, and the field of view angles are 18° and −18°, respectively, and the average error of measurement is increased when compared with area A; Areas D and E are close to the upper and lower edges of the field of view, and the vertical field of view angles are −18° and 18°, respectively. The average error of measurement is larger than that of area

A, and it is also obviously larger than that of areas B and C; the measurement error of area F is slightly larger than that of area A.

The above results show that the reconstruction accuracy of the central region is the highest in the field of view of the binocular reconstruction system, and the reconstruction error changes little with the Z coordinate value. In the area with a large horizontal or vertical field of view angle, the reconstruction error increases obviously, and the influence of the vertical field of view angle on accuracy is greater than that of the horizontal field of view angle. The above measurement results are consistent with the theoretical analysis in "The Derivation and Analysis of Structural Parameters based on Simulation".

## DISCUSSION AND CONCLUSION

This article aims to study the influence of structural parameters on measurement errors in a binocular 3D reconstruction system and to provide a reasonable range of values for each structural parameter since the binocular 3D reconstruction technology is the closest implementation method to human vision with more stable algorithms and easier implementation characteristics and is widely used in technical fields such as industrial measurement and virtual reality with immense development potential. Moreover, this article aims to establish an analytical relationship between structural parameters and error propagation coefficients using the analytical model. The ideal ranges for structural parameters guided by reducing error propagation coefficients are obtained.

The measurement accuracy of a binocular 3D reconstruction is the most important technical indicator, so there have been many studies on high-precision binocular 3D reconstruction in the literature. Generally, algorithms mainly focus on camera calibration and feature recognition. However, structural features should be counted too since structural deformation errors may occur due to structural changes caused by gravity, inertial forces, vibration, and temperature changes.

The main structural parameters include the length of the binocular baseline, the angle between the optical axis of the binocular camera and the baseline, and the lens focal length. The literature shows that the influence of structural parameters on error measurements is one of the most important factors among many influencing factors. Therefore, research shows that the alteration of structural parameters could improve the resolution and measurement accuracy of the system.

A binocular 3D reconstruction model is established based on the principle of triangulation, and the coordinate expression of the target point is established using analytical methods. Then, the theoretical expression of the error measurement propagation coefficient for the system is derived. Experiments verify both theoretical analysis and simulations.

Binocular reconstruction has technical advantages such as simple principles and calculations, easy realization, and higher accuracy. At the same time, binocular reconstruction is also the basic component of multi-view reconstruction. In this article, the basic composition and principle of the binocular reconstruction system are studied. Moreover, the structural parameters and error analysis model of the binocular reconstruction system are established. The analytical calculation method of a spatial point

position is presented based on geometric optics, and the influence of structural parameters on the error propagation coefficient of the reconstruction system is further deduced and analyzed.

An experimental platform was built based on the results of the theoretical analysis. The collected landmark displacements were compared with those of the precision displacement table. The results showed that when the included angle between the optical axis and the baseline of the binocular reconstruction system was between 30° and 50°, and the horizontal and vertical field of view angles were close to 0°, or reducing the ratio of the baseline length to the focal length by appropriately, the error propagation coefficient of the reconstruction system could be reduced. Structural parameters that affected the error propagation coefficient in theoretical analysis and simulation are verified. A basis for building a binocular reconstruction system with higher precision is suggested.

More specifically, we run experiments to determine the error propagation coefficients influenced by the relationships between the binocular optical axis and the baseline and the horizontal field of view, and the vertical field of view. The simulation data shows that the values of the included angles $\alpha 1$ and $\alpha 2$ are significant. The error propagation coefficient has a significant upward trend as the two included angles increase. The error propagation coefficient is smaller than at locations where the two included angles are unequal. So, the angle between the binocular optical axis and the baseline should be as close as possible to reduce the error propagation coefficient.

The simulation data shows that when the angle between the optical axis and the baseline is within the range close to 0°, the error propagation coefficient presents an increasing trend since the angle of the optical axis decrease. While the angle changes between 25° to 55°, the error propagation coefficient would be small. However, when the angle is greater than 60°, the error propagation coefficient increases sharply. The angle between the optical axis of the binocular camera and the baseline should be between 25° and 60° to reduce the error propagation coefficient is suggested.

When other conditions are constant, reducing the baseline length or selecting a lens with a long focal length could reduce the measurement error and the error propagation coefficient is inversely proportional to the focal length, so choosing a lens with a long focal length could reduce the error propagation coefficient. In addition, the lens with a long focal length should be selected as far as possible under the condition that meets the field of view, which could thus reduce the reconstruction error.

When the angle between the binocular optical axis and the baseline changes 30° through 60°, the error propagation coefficient would be in a lower range, which could reduce the measurement error caused by calibration error.

The influence of structural parameters on measurement errors in a binocular 3D reconstruction system is investigated with a limited number of attributes. The number of attributes would be increased in future research to reach a more complete binocular 3D reconstruction system.

### Funding
The authors received no funding for this work.

### Competing Interests
The authors declare that they have no competing interests.

### Author Contributions
- Ou Sha conceived and designed the experiments, performed the experiments, performed the computation work, authored or reviewed drafts of the article, and approved the final draft.
- Hongyu Zhang conceived and designed the experiments, performed the experiments, analyzed the data, prepared figures and/or tables, authored or reviewed drafts of the article, and approved the final draft.
- Jing Bai conceived and designed the experiments, prepared figures and/or tables, authored or reviewed drafts of the article, and approved the final draft.
- Yaoyu Zhang conceived and designed the experiments, performed the experiments, analyzed the data, performed the computation work, authored or reviewed drafts of the article, and approved the final draft.
- Jianbai Yang conceived and designed the experiments, performed the experiments, analyzed the data, performed the computation work, prepared figures and/or tables, authored or reviewed drafts of the article, and approved the final draft.

### Data Availability
The code is available in the Supplemental File. The data is available at ModelNet: https://paperswithcode.com/dataset/modelnet.

### Supplemental Information
Supplemental information for this article can be found online at http://dx.doi.org/10.7717/peerj-cs.1610#supplemental-information.

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
