# Peer review of "The analysis of the structural parameter influences on measurement errors in a binocular 3D reconstruction system: a portable 3D system"

_PeerJ Computer Science, doi:10.7717/peerj-cs.1610_

## Round 0.1 · original submission · Major Revisions

Please see the reviewers' detailed feedback. The authors should address concerns, improve the clarity and coherence of the content, and consider adding a discussion section to elaborate on the results and their implications. Reviewers have pointed out that the methodology section lacks clarity regarding specific algorithms and methodologies used for error analysis and simulation. The choice of experimental parameters and objects needs clarification, and claims in the experimentation section require validation. Additionally, there are formatting errors, unclear definitions of terms, and equations that lack explanations or citations.

Reviewer 1 ·

Basic reporting

No comment

Experimental design

No comment

Validity of the findings

No comment

Additional comments

Overall this paper is written well and contributes technically to the field of 3D reconstruction system . I have following suggestions which can improve the quality of the paper.
1. Title may be improved as there are multiple terms for a better understanding of research problem and potential contribution
2. Abstract is not well written; it does not express the research problem well, moreover the contribution points of this work may be further improved.
3. Overall manuscript has multiple grammatical mistakes, typos, unnecessary spaces etc.
4. The entire manuscript is not well-structured; a de facto structuring style may be followed for such scientific papers, i.e., Abstract, Introduction, Literature Review, Methodology, Results and Discussion and conclusion
5. Many terminologies need a minute explanation for the reader, and many abbreviations are used without complete form.
6. The introduction section is well written, but the first three sentences need a few more sentences about types of 3D reconstructions before jumping into a binocular 3D reconstruction
7. The third paragraph of the Introduction section defines the objectives of this paper; it should mention the list of coefficients under consideration.
8. Author must indicate structural parameters improved in the abstract and discussion sections
9. The “principle of triangulation” is used but neither defined nor cited.
10. The paper is related to 3D images, but most of the coordinates are 2D, i.e., (x,y) in most of the manuscript either it’s a typo, or it should be discussed.

Reviewer 2 ·

Basic reporting

Unusual spaces in section 2, some formatting errors at multiple lines
Define terms ∆L, ∆α1, ∆α2, ∆f1, ∆f2
Add a few more references in the experimentation section since most of the equations need citations
Not even a single equation was cited
Equations 2,3 and 4 needs explanations, i.e., l,h,f,∝
Term formulas and Equations are used alternatively, which is misleading; at some point, authors referred to formula (1) and another point as equation (12).
All those derivatives in equations 12 and 13 need explanations or refer to the source of the work
Why two influences, i.e., angle and horizontal fields, were taken and why particular objects for experimentations?
The following two sentences in the experimentation section need validation of the claim; if it is a contribution, it must be mentioned in the abstract and conclusion paragraph. “”
Angle 30 degrees and 50 degrees shows good results OR from 30 to 50 shows good results, must be cleared since it is miss leading
Conclusion of the manuscript is not coherent with the experimentation
Change the title of Conclusion and Discussion to discuss results and how they are improving the existing state of the art. Or add a new section Discussion may be added
The last sentence of the paper consists of 97 words, such wordy sentences often mislead the reader

Experimental design

See above

Validity of the findings

See above

Additional comments

no further comments

·

Basic reporting

Overall, the paper presents an interesting study on the influence of structural parameters on the measurement errors in a binocular 3D reconstruction system. The topic is relevant and significant in the field of computer vision, particularly in the context of high-precision 3D reconstruction. However, there are some concerns and areas that need improvement before the paper can be considered for publication:
• Lack of Clear Objectives: The paper lacks a clear and well-defined research question or objective. It is essential to explicitly state the main goals and contributions of the study in the introduction section to guide the readers throughout the paper.
• Inadequate Literature Review: The literature review provided is limited and does not sufficiently cover the existing research on the influence of structural parameters on measurement errors in binocular 3D reconstruction. More comprehensive and up-to-date references are required to provide a stronger background for the study.

Experimental design

• Methodological Rigor: While the authors mention using MATLAB for simulations, there is insufficient detail regarding the specific methodologies and algorithms employed for the error analysis and simulation. The lack of clarity in the methodology section hampers the reproducibility of the study.
• Experimental Validation: The paper mentions the conduction of experiments to verify the theoretical calculations and simulation analyses. However, the experimental setup, data collection process, and the results of these experiments are not adequately described. Without this information, it is challenging to assess the validity and reliability of the experimental findings.

Validity of the findings

• Missing Results Discussion: The conclusions provided in the paper lack a comprehensive discussion of the results and their implications. The authors should elaborate on the significance of the observed error reduction when the optical axis and baseline angle are between 30° and 50° or when the baseline length to focal length ratio is appropriately reduced. Moreover, the authors should relate their findings to the existing literature and discuss potential practical applications of their research.

Additional comments

• Clarity and Structure: The overall structure of the paper could be improved to enhance readability. The flow of information between sections needs to be better organized, and the subsections should have clear and informative titles.
• Limitations and Future Work: The paper does not adequately address the limitations of the study and potential avenues for future research. The authors should acknowledge the shortcomings of their work and suggest possible directions for further investigation to enhance the impact of their findings.

In conclusion, while the study addresses an essential aspect of binocular 3D reconstruction, it requires substantial revisions to meet the standards of a rigorous and well-documented scientific publication. Addressing the issues mentioned above would strengthen the paper and provide valuable insights to the research community interested in high-precision 3D reconstruction systems.

---

## Round 0.2 · accepted · Accept

All reviewers have confirmed that the authors have addressed the comments.

Reviewer 1 ·

Basic reporting

No comment

Experimental design

No comment

Validity of the findings

No comment

Additional comments

The authors have satisfactorily resolved all of my concerns. Therefore, I suggest this article for publication in its current form.

Reviewer 2 ·

Basic reporting

overall, the paper is well improved according to my previous comments and therefore, no more comments

Experimental design

The revised version of the paper contains validated experimental design

Validity of the findings

validities are of good and acceptable quality

Additional comments

i am satisfied with the current version of the paper

·

Basic reporting

Authors addressed all my concerned in the revised version of the manuscript.

Experimental design

Authors addressed all my concerned in the revised version of the manuscript.

Validity of the findings

Authors addressed all my concerned in the revised version of the manuscript.

Additional comments

Authors addressed all my concerned in the revised version of the manuscript.